# New Insights into Dyskerin-CypA Interaction: Implications for X-Linked Dyskeratosis Congenita and Beyond

**DOI:** 10.3390/genes14091766

**Published:** 2023-09-06

**Authors:** Valentina Belli, Daniela Maiello, Concetta Di Lorenzo, Maria Furia, Rosario Vicidomini, Mimmo Turano

**Affiliations:** 1Istituto Nazionale Tumori—IRCSS—Fondazione G. Pascale, 80131 Naples, Italy; valentina.belli@istitutotumori.na.it; 2Department of Biology, University of Naples Federico II, 80126 Naples, Italy; daniela.maiello@unina.it (D.M.); concetta.dilorenzo@unicampania.it (C.D.L.); mfuria@unina.it (M.F.); 3Department of Advanced Medical and Surgical Sciences, University of Campania “Luigi Vanvitelli”, 80138 Naples, Italy; 4Section on Cellular Communication, Eunice Kennedy Shriver National Institute of Child Health and Human Development (NICHD), National Institutes of Health (NIH), Bethesda, MD 20892, USA

**Keywords:** PPIase A, proline isomerization, H/ACA RNPs, dyskerin, DKC1, X-DC, ribosomopathies, redox response

## Abstract

The highly conserved family of cyclophilins comprises multifunctional chaperones that interact with proteins and RNAs, facilitating the dynamic assembly of multimolecular complexes involved in various cellular processes. Cyclophilin A (CypA), the predominant member of this family, exhibits peptidyl–prolyl cis–trans isomerase activity. This enzymatic function aids with the folding and activation of protein structures and often serves as a molecular regulatory switch for large multimolecular complexes, ensuring appropriate inter- and intra-molecular interactions. Here, we investigated the involvement of CypA in the nucleus, where it plays a crucial role in supporting the assembly and trafficking of heterogeneous ribonucleoproteins (RNPs). We reveal that CypA is enriched in the nucleolus, where it colocalizes with the pseudouridine synthase dyskerin, the catalytic component of the multifunctional H/ACA RNPs involved in the modification of cellular RNAs and telomere stability. We show that dyskerin, whose mutations cause the X-linked dyskeratosis (X-DC) and the Hoyeraal–Hreidarsson congenital ribosomopathies, can directly interact with CypA. These findings, together with the remark that substitution of four dyskerin prolines are known to cause X-DC pathogenic mutations, lead us to indicate this protein as a CypA client. The data presented here suggest that this chaperone can modulate dyskerin activity influencing all its partecipated RNPs.

## 1. Introduction

Cyclophilins are highly conserved proteins known for their molecular chaperone activity, which facilitate protein folding and trafficking [1]. Peptidyl–prolyl cis–trans isomerase A (PPIase A), also named cyclophilin A (CypA), the most abundant member of this family, is widely recognized for its intracellular role and is a major target of the immunosuppressive drug cyclosporin [2] that is widely used for the treatment of several tumors. Beyond its involvement in immune modulation, CypA is implicated in various cellular processes and has been associated with numerous pathological conditions, including aging, stress, cancer, cardiovascular diseases, and neurodegeneration [3]. Founded on its ability to also bind to RNA, the protein plays additional roles in the progress of viral infections [4] and is needed for the replication of several coronaviruses, including SARS-CoV19 [5,6]. While CypA is distributed between the cytoplasm and the nucleus [7], its nuclear functions remain to be fully explored. However, studies have shown that CypA plays a crucial role in regulating the composition and architecture of different heterogeneous nuclear ribonucleoprotein (RNP) complexes, which are involved in chromatin remodeling, RNA dynamics, and nuclear receptor signaling [7]. This regulatory role is further enhanced by the fact that, as with other cyclophilins, CypA is endowed with a peptidyl–prolyl cis–trans isomerase (PPIase) activity that is known to give an additional contribution to protein conformational stability. Peptidyl–prolyl isomerases play important roles not only in protein trafficking and secretion, but also in the assembly of large complexes, by favouring proper inter/intra-molecular interactions [8,9]. 

Unlike most chaperones, PPIase are specific in their respective targets: they preferentially bind proteins that contain conformationally heterogeneous proline residues, whose conformation change often acts as a switch in the regulation of specific functions [10]. Such types of configuration changes are known to be essential to the dynamic influx–outflux of the various components of membrane-less (ML) foci, biological condensates that coalesce through multiple transient interactions and act as compartmentalized functional hubs within the cell [11]. Thanks to the lack of an external membrane, ML bodies assume a liquid droplet-like behaviour, so that their components can continuously be interchanged and their structure and organization rapidly rearranged [12]. 

Proteins harbouring Intrinsically Disordered Regions (IDRs) are known to have a relevant role in the assembly of these foci [13,14], since IDRs can mediate transient assembly with different interacting partners and promote Liquid–Liquid Phase Separation (LLPS), that is widely recognized as the main mechanism driving the coalescence of ML bodies [15,16]. Inside the nucleus, many ML bodies act as specialized subcompartments composed of locally concentrated RNA-protein assemblies where specific functions take place. The nuclear membrane-less (NML) bodies ensemble includes primarily nucleoli, nuclear speckles, paraspeckles, PML, Sam68, and Cajal bodies [17]. While an increasing number of these NML foci are continuously revealed by modern imaging/molecular techniques, the nucleoli still appear to be those playing the most prominent cellular roles. In addition to representing the milieu where ribosomes assemble their molecular components, these multifunctional organelles act as hubs in the coordination of many cellular responses, being able to dynamically change structure and composition in response to various stimuli, such as growth conditions, stress, DNA damages, signaling events, and viral infections [18]. Many key nucleolar proteins are known to contain IDRs, and their dysregulation proved to impact the LLPS maintenance of the nucleolar structure [19]. Given that IDRs are particularly rich in prolines [14], PPIase activity emerges as a post-translational modification that can modulate LLPS [11]. Notably, CypA has been identified as a client protein of the Upstream Binding Transcription Factor (UBTF) and other key nucleolar proteins, suggesting its major role in LLPS-based nucleolar organization [20]. This notion is supported by the close connection between rDNA transcription, ribosome biogenesis, and the initial steps of nucleolar phase separation [21]. This view is well consistent with the observation that rDNA transcription, together with the initial steps of ribosome biogenesis, is connected to the primary events of nucleolar phase-separation [21] and is further supported by recent NMR spectroscopic studies showing that CypA plays a regulatory role in the phase organization of droplets formed by proteins harboring proline-rich IDRs [22]. As is well known, the nucleolus has a liquid droplet tripartite structure [23,24] composed of the internal fibrillar center (FC), containing rDNA chromatin, the surrounding dense fibrillar component (DFC), where most processes important for transcription, processing, and modification of ribosomal RNAs take place, and the external granular component (GC) containing the immature ribosomes [25]. 

Continuous trafficking occurs not only between the nucleolus and the surrounding nuclear regions, but also within its tri-layered body. This dynamic interchange is assisted by multiple resident chaperones and co-chaperones that act by favoring the structural arrangement of large macromolecular aggregates, allowing protein–protein and protein–nucleic acid interactions to be continuously renewed and reestablished. 

Here, we indicate CypA as one of the chaperones involved in these nucleolar dynamics. We report that this protein can localize within the nucleoli of human cell lines and that it interacts with dyskerin (UniProt # O60832), a highly conserved nucleolar protein belonging to the family of RNA-guided pseudouridine synthases. Dyskerin, whose sequence includes several IDRs [26], is a multifunctional RNA-binding protein that, in association with the highly conserved GAR1, NHP2, and NOP10 proteins, enters in the composition of the H/ACA RNPs active tetramer [26,27]. Within the nuclei, dyskerin concentrates at the nucleoli and the Cajal bodies (CBs), two intimately connected NMLs bodies [28]. At these two sites reside or transit the main dyskerin-participated functional complexes: the telomerase holoenzyme and the small Cajal body RNPs (scaRNPs), specifically involved in the modification of spliceosomal snRNAs and the small nucleolar RNPs (snoRNPs) that control ribosome biogenesis and RNA pseudouridylation [27]. By elucidating the interaction between CypA and this multi-faced pseudouridine synthase, our data suggest that CypA assists in the assembly, organization, and trafficking of dyskerin-participated RNPs. This discovery not only expands our understanding of the repertoire of CypA functions, but also holds clinical significance. In fact, mutations of dyskerin, encoded by the *DKC1* gene (OMIM: #305000), cause two extremely pleiotropic congenital ribosomopathies: the X-linked dyskeratosis (X-DC) [29] and the more severe Hoyeraal–Hreidarsson syndrome (HHS) [30], while the protein over-expression characterizes several types of sporadic cancers and is valued as a marker of poor prognosis [31]. 

## 2. Materials and Methods

### 2.1. Cell Culture, Treatments and Bioinformatic Data 

RKO, HEK, and U2OS cell lines were maintained at 37 °C in a humidified atmosphere of 5% CO_2_ in Dulbecco Modified Eagle Medium (DMEM) containing 10% Tet-free fetal bovine serum (FBS), 1% penicillin-streptomycin, and 1% L-Glutamine (Euroclone, Milano, Italy). To generate HEK 293T and U2OS (ATCC, Manassas, VA, USA) transfected cells, 3 µg of pMGIB-3xFLAG-dyskerin (Addgene) and 12 µL of Polyethylenimine (PEI) (Polysciences) were added to the medium; following 30 min of incubation, cells were put on plate for 5 h; the medium was then replaced by DMEM supplemented with 20% FBS. Heat shock was performed by exposing cells at 45 °C for 30 min; cells were allowed to recover for 30 min at 37 °C and then analyzed by immunofluorescence or immune precipitation as described below.

HeLa cells over-expressing dyskerin isoform 3 (3XF-Iso3) were previously obtained [32] and cultured and subjected to sodium arsenite treatment as described [33]. Unless otherwise specified, all published protein interactions were retrieved from the BioGrid database version 4.4.223 (http://thebiogrid.org (accessed on 7 July 2023)).

### 2.2. Immunofluorescence Analysis 

For immunofluorescence experiments, cells were directly seeded onto glass coverslips then permeabilized and exposed to antibodies as previously described [33]. Immunofluorescence micrographs were taken by Zeiss LSM 700 microscope (Zeiss, Oberkochen, Germany) using EC Plan-Neofluar 40× or Plan-Apochromat 63x/1.40 Oil immersion objectives. Images were analyzed by ImageJ (http://imagej.nih.govt/ij/ (accessed on 5 June 2023; 21 August 2023)) or Zeiss Zen software tools. Nuclei were counterstained by DAPI. The primary antibodies used were anti-DKC1 H-3 (SC-373956), Santa Cruz Biotechnology, Santa Cruz, CA, USA; anti-CYPA/PPIA (Genetex 104698); anti-FLAG (F1804–200UG, Sigma-Aldrich, St. Louis, MO, USA); and anti-PRDXII (Ab133481, ABCAM). The secondary antibodies used were anti-Rabbit-Cy3 (A10523) and anti-Mouse-AlexaFluor 488(A21202) from Invitrogen and anti-Rabbit-Cy3 (A120-201C3), anti-Rabbit-FITC (A120-101F), anti-Goat-Cy3 (A50-201C), anti-Mouse-FITC (A090-138F), and anti-Mouse-Cy3 (A90-516C3) from Bethyl Laboratories, Montgomery, AL, USA. Nuclei were counterstained by DAPI. Each experiment was repeated at least three times independently.

### 2.3. Pull-Down ASSAY 

In these assays, HEK and U2OS cells were transfected for 48 h with 3xFLAG-dyskerin plasmid (Addgene), as described above and then transferred in RIPA lysis buffer (50 mM Tris pH 8, 150 mM NaCl, 5 mM EDTA, 1% NP40) supplemented with anti-phosphatase and anti-protease (cOmplete™, Mini, EDTA-free Protease Inhibitor Cocktail, Sigma-Aldrich, Merk, Rahway, NJ, USA). Lysates of transfected cells (about 1 mg) were added to 20 μL of TBS 1x-rinsed beads (ANTI-FLAG M2 Affinity Gel A2220-SIGMA ALDRICH) and incubated overnight at 4 °C. Following a 1 min spin at 8000× *g* at 4 °C, unbound proteins were removed by rinsing beads with 200 μL of TBS 1x wash buffer three times. Pulled-down proteins were then analyzed by immunoblotting. The experiments were repeated three times independently. The primary antibodies used were anti-CYPA/PPIA (104698, Genetex); anti-PRDXII (Ab133481, ABCAM); anti-Zo-1 (bs1329, Bioss); anti-GAR1 (PA5-63656, Thermofisher); and anti-FLAG (F1804–200UG, Sigma-Aldrich, St. Louis, MO, USA). Secondary antibodies were anti-Rabbit (A120-100P) and anti-Mouse (A90-137P), Bethyl Laboratories, Montgomery, AL, USA.

## 3. Results

### 3.1. CypA Can Colocalize with the Dyskerin Pseudouridine Synthase in the Nucleoli

Several proteins with various intracellular localizations play different roles at the distinct subcellular locations. To provide new insight into the roles played by CypA within the nucleus, we conducted a detailed analysis of its intracellular localization using confocal immunofluorescence microscopy (IF). When cells from the RKO, U2OS, and HeLa lines were examined, we found that, although in each line CypA appeared to be widely diffused throughout the whole cell bodies, its nuclear pattern showed some cell-type specificity. In fact, in the RKO and U2OS cells, the protein’s immunosignal marked intensely the nucleoplasm, where many positive puncta were detected, whereas in the HeLa cells the nucleoplasm remained nearly unlabelled (Figure 1A–C). Despite these cell-type differences, in each line a fraction of the CypA immunosignal concentrated within the nuclei at main bodies that, as shown by their colocalization with the nucleolar marker dyskerin, corresponded to the nucleoli (Figure 1).

Focusing on CypA-dyskerin co-stained cells, we noted that the CypA’s immunosignal, although generally concentrated in the fibrillar center, also labelled peripheral foci, suggest a lively transit throughout the nucleolar tri-layered organization (Figure 1). The more pronounced peri-nucleolar localization of CypA in HeLa cells as compared to RKO and U2OS cells, as depicted in Figure 1, may indeed reflect the complex interplay between cellular context, metabolic conditions, and protein dynamics. 

One plausible explanation could be attributed to the inherent heterogeneity in cellular environments, such as differences in organelle composition, signaling pathways, and gene expression profiles. It is well-known that different cell types may exhibit unique subcellular distributions of various proteins, which can arise from variations in their functional roles and regulatory mechanisms.

Furthermore, post-translational modifications and interactions with partner molecules can also significantly influence protein localization. Given that dyskerin is also highly concentrated at CBs, where the assembly of the dyskerin-partecipated telomerase holoenzyme initiates and the mature H/ACA SnoRNPs are thought to be gathered [34], we also looked at these sites. CBs shape as a consequence of an elevate concentration of both snRPs and snoRNPs and, not being essential, may even be undetectable in the same cells [35]. In our experiments, putative CBs, labelled by the dyskerin antibody, were hardly visible within the small nuclei of RKO cells, instead being well evident in the U2OS and HeLa cells. Yet, in both HeLa and U2OS cells the CypA immunosignals still partially overlap at these sites (Figure 1A’,B’, white arrows). Based on the different intensity at which the CypA immunosignal was simultaneously detected in the same cells (Figure 1), the nucleoli should be expected to represent the major sites where CypA concentrates. Indeed, it is plausible that the lowering of CypA concentration might favour the locally assembled mature H/ACA RNPs in stabilizing their active conformational structures. To this regard, we noted that another cyclophilin member, CypB, was reported to localize at the nucleoli of the HeLa and human primary fibroblast cells, whereas it remained undetected at CBs in normal growth conditions [36]. Thus, it is possible that a specific major nucleolar concentration might represent a common pattern for nuclear clyclophilins.

In addition to its potential involvement in dyskerin-RNP dynamics, we speculate that the nucleolar concentration of CypA might have, per se, a functional relevance for the redox response of this organelle [37]. In fact, although CypA participates in the maintenance of cell homeostasis in several ways, a significant contribution is given by its ability to act as an endogenous electron donor. With its reducing action, the protein is known to actively support the antioxidant activity of mammalian peroxiredoxins (PRDXs), being able to bind them and help to restore their redox activity [38,39]. Interestingly, PRDX-2, a relevant member of this ROS scavenger family, has been found to localize at the nucleoli [40,41]; (Figure 2). This suggests that, under certain conditions, CypA and PRDX-2 may act together as localized ROS sensors, thereby contributing to the orchestration of the nucleolar antioxidant response. It is worth noting that dyskerin has also been implicated in the modulation of oxidative metabolism, further connecting it to these non-canonical functions [35,40,42]. Consistent with this, by analyzing the fluorescence intensity of dyskerin and PRDX-2 immunoreactivities along a rectangular region of interest (ROI) containing the nucleoli, it became evident that the peaks of PRDX-2 fluorescence significantly overlap with the peaks of the dyskerin signal, suggesting a potential cooperation between the two proteins in redox-related cellular processes [Figure 2].

### 3.2. Dyskerin as a Putative Novel CypA Client 

Since CypA has been correlated to specific nucleolar functions, such as rDNA transcription and ribosome biogenesis [20,21], we investigated the possibility that it could directly interface with dyskerin. To explore this, we transfected both HEK and U2OS cells with Flag-tagged full-length dyskerin and tested the obtained lysates with the anti-FLAG^®^ M2 Magnetic Beads in a dyskerin pull-down assay (see Methods). As shown in Figure 3A, CypA was effectively co-pulled down with the Flag-tagged dyskerin; it was also recovered in the immunoprecipitate after a mild heat shock exposure, indicating that this condition did not disrupt this interaction. In these experiments the protein GAR1, a core component of the H/ACA snoRNP mature complex, was also co-pulled, providing a positive control; a suitable negative control was instead furnished by the absence in the pulled immunocomplex of the membrane protein Zo-1 (Figure 3A,B). 

To further investigate the interfacing of CypA with dyskerin, we extended IF analyses to the low abundance dyskerin splice isoform 3 (thereafter called Iso3), whose expression is characterized by a different intracellular pattern. This truncated protein, 420 aa long, maintains all dyskerin functional domains with exception of the C-terminal NLS, whose absence allows a variegated dual cytoplasmic–nuclear localization; nevertheless, within the nuclei, Iso3 concentrates into nucleoli and CBs, similarly to canonical dyskerin [32,33,43]. Like dyskerin [44,45,46], this isoform is actively involved in the stress/DNA damage response [33,47] and, upon oxidative stress, is efficiently recruited into stress granules (SGs), as components of mature H/ACA snoRNPs [33]. SGs are ML cytoplasmic bodies that promote adaptation to stress conditions and cell survival [48], whose nucleating factor is the RNA-binding protein Ras-GTPase-activating protein (G3BP1) [49,50]. To follow the expression of CypA and Iso3 in the absence or presence of oxidative stress, we subjected the previously obtained stably transfected Hela cell lines that overexpress the FLAG-tagged Iso3 (3XF-Iso3 cells) to IF analysis [32]. Strikingly, in the unstressed condition, Iso3 (marked by the FLAG antibody) and CypA immunosignals were found to colocalize not only into the nucleoli, but also in the cytoplasm (Figure 3C). This indicates that the truncated dyskerin variant includes all sequences eventually required for the interaction with CypA, and that the interfacing is not restricted to the nuclei. Given that the canonical dyskerin, although in trace amounts, also resides in the cytoplasm [33,51] this issue suggested that CypA might have a chaperoning role for both newly synthesized isoforms, possibly contributing to their immediate stabilization and/or preventing them from unspecific and incorrect interaction isoforms also in this cell compartment. Considering the presence of two nuclear localization signals (NLS) in dyskerin, it is also plausible that CypA might translocate to the nucleus through an interaction with this protein. Next, we exposed 3XF-Iso3 cells to sodium arsenite, which is known to induce oxidative stress and is widely used to trigger SGs formation. In this condition not only Iso3, but also CypA coalesced into SGs, as previously noticed by other authors [9,52]. As shown in Figure 3D, the CypA’s immunosignal overlapped with that of Iso3 at a subset of SGs, indicating that co-localization of these two proteins persisted upon oxidative stress. It is plausible that the number of Iso3-positive SGs not marked by CypA might be due to the high level of Iso3 overexpression in the 3XF-Iso3 cells and/or from the general structural plasticity of SGs, whose individual components are known to continuously shuttle from them to the cytoplasm or to other cytoplasmic bodies [53]. It is worth noting that the coalescence of CypA into SGs fits well not only with its role of molecular chaperone in the refolding of denatured proteins following oxidative stress [7], but also with the knowledge that many proteins that have a nucleolar localization are recruited into these bodies in response to stress stimuli [54]. Finally, it adds functional significance to the remark that G3BP1 is a dyskerin-CypA common interactor (as listed at the www.thebiogrid.org (accessed on 7 July 2023), version 4.4.223). Considering the evidence supporting the close association between CypA and dyskerin, we hypothesized that the peptidyl–prolyl isomerase (PPLase) activity of CypA might specifically target dyskerin at certain proline residues. Structural information on eukaryotic H/ACA RNPs is still uncomplete, so we focused on the analysis of sequence conservation and the mutational hot spots of dyskerin protein sequences. As shown in Figure 3E, dyskerin harbors several highly conserved proline residues, most of which are embedded within its main functional domains: the dyskerin-like domain (DKLD; 48-106 aa), whose function remains so far elusive and the TruB pseudouridine synthase domain (110-226 aa), which acts catalytically in the RNA pseudouridylation process and the PUA RNA binding domain (297-370 aa), which allows RNA recognition [55]. Dyskerin-prevalent nuclear localization is also ensured by two Nuclear Localization Signals (NLSs) localized, respectively, at the N- and C-ends. Specifically, five conserved proline residues lie within the N-terminal moiety that includes the DCLD domain, three in the TRUB catalytic domain, one in the RNA-binding PUA domain, and five in the subsequent C-terminal moiety (Figure 3E). While these prolines are distributed across various domains of dyskerin, it is plausible that the interaction between CypA and dyskerin may involve specific structural motifs and conformational changes rather than a uniform recognition of all proline residues. Considering that structural information on eukaryotic H/ACA RNPs is still uncomplete, we next focused on the analysis of sequence conservation and the mutational hot spots of dyskerin protein sequence. We next compared the position of these prolines with the distribution of the known X-DC pathogenic missense mutations and remarked that they overlap each other at four codons (Pro10Leu, Pro40Arg, Pro384Ser, Pro384Leu, Pro409Leu, Pro409Arg, all labelled by an asterisk in Figure 3E). Substitutions of these four prolines are known to affect dyskerin functions [30,56], in good accordance with the possibility that they might represent putative CypA targets. The observation that these codons all map within the sequence of the truncated Iso3 isoform that proved to retain the ability to interact with CypA further supports this hypothesis. Noticeably, two different single nucleotide substitutions fall at two of these codons (Pro384 and Pro409), giving rise to two diverse pathogenic missense mutations at each site (Pro384Ser, Pro384Leu, Pro409Leu, Pro409Arg). This raises the possibility that the substitution of Pro384 and Pro409 might potentially play relevant roles in dyskerin activity. Indeed, these two residues map within the PUA downstream flanking region, which is highly conserved among eukaryotes and is required for the interaction with SHQ1, the assembly factor involved in the early steps of H/ACA RNP biogenesis [57]. Although direct evidence is still lacking, the data presented here strongly support the hypothesis that the PPLase activity of CypA might specifically target dyskerin, potentially modulating its activity and H/ACA RNPs dynamics, in line with the role attributed to CypA on the establishment of the LLPS-based nucleolar architecture.

### 3.3. Overlapping of CypA, Dyskerin and Fibrillarin Interactomes

H/ACA and C/D box SnoRNPs are the major resident nucleolar complexes involved in rRNA maturation and processing, with dyskerin and fibrillarin representing their respective catalytic core members and those that primarily contact RNAs [29]. Both RNP families are assembled throughout dynamic multistep pathways characterized by a lively intranuclear trafficking between nucleoli and CBs [29,58]. These interchanges rely on transient interactions and are regulated by a diverse set of chaperones [29,58,59]. Given the involvement of chaperones in these processes, we explored the potential role of CypA in assisting the dynamics of both RNP families, possibly during transient assembly events. To this aim, we analyzed the human interactomes of CypA, as well as the other H/ACA RNPs core components at the BioGRID Database of Protein Interactions. These RNPs include, in addition to dyskerin, the highly conserved NOP10 and NHP2 proteins; a fourth protein, NAF1, is transiently recruited at the site of transcription to be subsequently replaced by GAR1 [58]. After this exchange that is thought to occur at CBs [34], the mature complexes move back to the nucleolus, where they are actively involved in the processing and the modification of rRNA. 

First, we compared the human interactomes of CypA, dyskerin, and fibrillarin. This comparison showed that a significant fraction of fibrillarin (72 interactors) and dyskerin (49 interactors) registered partners were shared with CypA (Figure 4A,B), suggesting that these proteins could participate in common assemblies. Remarkably, a consistent number of partners (i.e., 34 members) was shared by all three proteins, delineating a possible context of common interactions. 

Interestingly, we noted that the dysfunction of almost all these common partners (30/34; 88.23%), despite their variegated functional roles, was invariantly related to diseases (Table 1). 

This observation underscores the functional relevance of the relationship between CypA and the core components of both major RNP families and aligns with the broad involvement of CypA in various diseases, wherein it influences different pathophysiological pathways depending on the complexes it engages with [3]. Overall, this remark supported the possibility that CypA could participate, at least transitorily, in common gatherings with these complexes. 

Clusterization of interactors common to CypA, fibrillarin, and dyskerin by using the PANTHER classification system (http://www.pantherdb.org (accessed on 7 July 2023)) showed that the most abundant protein classes were represented by a protein modifying enzyme (11.8%), cytoskeletal protein (11.8%), and gene-specific transcriptional regulator (11.8%), together accounting for up 35.4% of the total (Figure 4B). 

Upon a thorough examination of the known interactors of each H/ACA RNPs core component, dyskerin, Nop10, NHP2, Gar1, and NAF1, which are transiently recruited at the site of snoRNA transcription [29,58], a remarkable observation emerged: most interactors shared a substantial number of common partners with CypA (Figure 4C). Of note, NAF1 and NOP10 displayed the least number of interactions with CypA, possibly due to their involvement in the early stages of RNP assembly. This intriguing finding implies that CypA might preferentially interact with mature and active H/ACA snoRNPs, suggesting a dynamic association that might play a pivotal role in this process.

## 4. Discussion

The nucleus serves as a crucial site for the diverse functional roles of CypA, particularly in regulating the assembly and trafficking of large RNP complexes with distinct functions [7]. In this study, we provide new insights into the functional repertoire of this multifunctional chaperone by demonstrating its localization in the nucleoli and its interaction with both the canonical and Iso3 truncated variants of human pseudouridine synthase dyskerin. These findings suggest a potential role for CypA in nucleolar dynamics, which is consistent with previous data from various systems. Notably, CypA has been shown to function as a ribosome biogenesis chaperone in filamentous fungi [60], and its specific involvement in H/ACA RNP dynamics has been observed in S. cerevisiae [61]. A previous study identified NHP2, a core component of H/ACA RNPs, as a client of CypA and demonstrated that the cis/trans isomerization of the highly conserved Pro83 residue influenced H/ACA RNP assembly. In our study, we present several lines of evidence suggesting that CypA may also target human dyskerin, indicating that proline isomerization should be considered among the various post-translational modifications described for this multifunctional protein. Dyskerin, in addition to harboring a significant number of phosphorylated residues [55], is extensively subjected to PARylation and SUMOylation [62,63], two types of post-translational modifications known to contribute to the biogenesis, trafficking, and subnuclear localization of the RNPs and have the ability to modulate liquid–liquid phase separation (LLPS) [64]. Future experiments will elucidate whether dyskerin is involved in the early stages of nucleolar phase separation, as has been established for several key nucleolar proteins such as fibrillarin [21] and GAR1 [18], which, together with dyskerin, form the active tetramer of H/ACA RNPs [27]. 

Addressing the CypA-dyskerin interaction, the observation that it persists and can be modulated in response to stress signals, even at cytoplasmic stress granules (SGs), highlights the functional potential of this interaction. This is particularly noteworthy considering that pseudouridylation affects the entire transcriptome and can be modulated by environmental signals [64]. 

Moreover, considering that the PPLase activity of CypA may play a critical role in influencing the local and flanking protein structures, it is plausible that this modification could impact the broad range of non-canonical dyskerin functions, which currently include redox metabolism [39], vesicular trafficking [65], and its ability to act as a co-transcriptional factor [66]. For that reason, the discovery of the CypA-dyskerin interaction holds significant relevance for patients affected by Dyskeratosis Congenita X-linked (X-DC). X-DC is a rare genetic disorder caused by mutations in the dyskerin-encoding gene *hDKC1*. Dyskerin is a key component of both telomerase and the snoRNA H/ACA complex [54]. Dyskerin functions as a pseudouridine synthase within the snoRNA H/ACA complex, guiding modifications in ribosomal RNA and other classes of RNAs essential for cellular processes. These modifications play important roles in RNA stability, function, and processing, contributing to vital cellular functions such as translation, pre-mRNA splicing, and ribosome biogenesis.

While telomerase dysfunction is central to X-DC, the pathogenesis of the disease is influenced by additional factors, including dyskerin telomerase-independent functions. The contribution of the telomerase-independent roles of dyskerin to the etiology of X-DC is further supported by studies on both animal and cellular models [39,45,65,67,68,69].

This multifaceted role highlights the complexity of X-DC, emphasizing the need to comprehensively understand dyskerin various cellular functions for potential therapeutic strategies. The identification of CypA-dyskerin interactions unveils an unknown dimension of dyskerin’s regulation, shedding light on potential disease-contributing pathways. Notably, this discovery holds several key points of relevance: 1. limited treatment options for X-DC necessitate new therapeutic targets. The interaction between dyskerin and CypA presents an avenue to explore modulating dyskerin’s activity to potentially alleviate symptoms or slow disease progression; 2. understanding how CypA influences dyskerin’s function may unravel the intricate regulation of pseudouridine activity, with implications beyond X-DC, including in cancer research; 3. as a member of the peptidyl–prolyl cis–trans isomerase (PPIase) family, CypA’s interaction with dyskerin suggests a role in ensuring proper folding and stability, contributing to a broader understanding of protein homeostasis and chaperone-mediated processes; 4. insights from this interaction may lead to specific biomarkers aiding in X-DC diagnosis and prognosis; 5. the interaction between dyskerin and CypA could explain phenotypic variations observed in affected individuals, informing personalized treatment approaches tailored to molecular defects; 6. the discovery opens avenues for therapeutic interventions targeting cellular senescence, stress response, and immune dysfunction associated with dyskeratosis congenita; and 7. dyskerin’s roles in stem cell maintenance and tissue regeneration may have implications for regenerative medicine in conditions where telomerase dysfunction affects tissue repair and renewal. 

This discovery enhances our understanding of dyskerin’s functions and offers potential opportunities to address the complexities of dyskeratosis Congenita X-linked and related cellular processes in various disease contexts.

## Figures and Tables

**Figure 1 genes-14-01766-f001:**
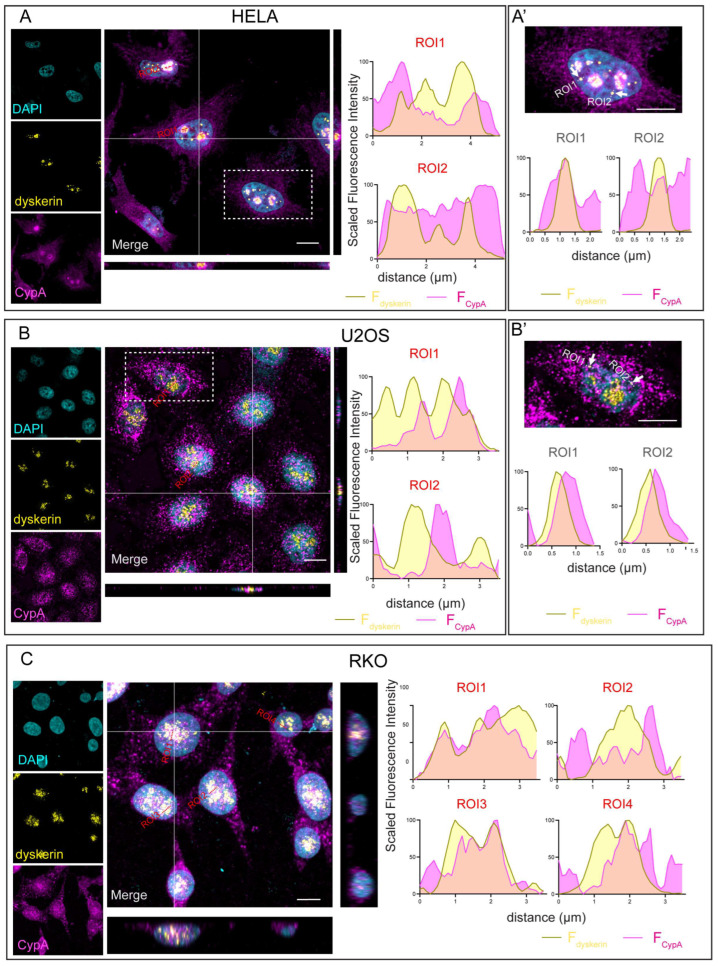
Intracellular Localization of CypA. Confocal microscopy images depicting distinct cell types, namely HeLa (**A**), U2OS (**B**), and RKO (**C**), immunostained with anti-dyskerin (yellow) and anti-CypA (magenta) antibodies, and counterstained with DAPI (cyan). In HeLa cells (**A**), as validated through Z-stack analysis (refer to XZ and YZ projections), the immunoreactive signal of CypA demonstrates a distribution spanning both the cytoplasm and the nucleus. Notably, fluorescence intensity profiles along two linear ROIs centered on nucleoli reveal CypA fluorescence exhibiting partial overlap with the immune signal of dyskerin (a nucleolar marker), indicating that CypA can assume a partial nucleolar localization. As expected, nuclear bodies (marked by white arrows), putatively corresponding to CBs, are also marked by dyskerin’s immune signal. (**A’**) Detailed analysis of fluorescence intensity profiles of the cell shown in the top enlargement (highlighted in (**A**)) along two linear ROIs centered on two putative CBs (ROI1 and ROI2—indicated in gray) illustrates CypA again surrounding and partially overlapping with dyskerin. (**B**) In U2OS cells, the distribution of the CypA signal similarly spans both the nucleus and the cytoplasm and, along two linear ROIs centered on nucleoli also exhibits a varying degree of overlap with nucleolar dyskerin signal. (**B’**) Here, again, analysis of fluorescence intensity profiles (cell on top enlargement, highlighted in (**B**)) shows that CypA and dyskerin signals are closely aligned and partially overlap along two linear ROIs (ROI1 and ROI2, (indicated in gray) centered on putative CBs (marked by white arrows in the magnified view). (**C**) Additionally, in RKO cells the immunoreactivity of CypA appears to be distributed between the cytoplasm and the nucleus. The analysis of fluorescence profiles along four linear ROIs (highlighted in red), each centered on a nucleolus, demonstrates a varying degree of overlap between the signals of CypA and dyskerin. Unlike the previous cell types, putative Cajal’s bodies are not clearly evident in this cellular context. ROI: Region of Interests; CBs: Cajal Bodies.

**Figure 2 genes-14-01766-f002:**
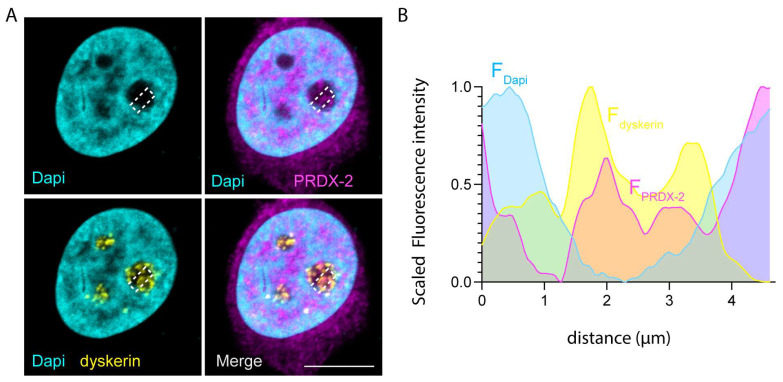
PRDX-2 intranuclear localization. (**A**) Confocal microscopy analyses of HeLa cells labelled with dyskerin (yellow) and PRDX-2 (magenta) antibodies and counterstained with DAPI (cyan). Note that PRDX-2 is widely diffused in the nucleoplasm and localizes within the nucleoli and the putative CBs. White bars correspond to 10 μm. (**B**) Graph showing the fluorescence intensity values for DAPI (cyan), dyskerin (yellow), and PRDX-2 (magenta), plotted as a function of distance along a linear ROI (indicated as a dotted rectangular outline in (**A**)). Dyskerin signal is observed within the nucleolus region, where the DAPI signal is decreased. A clear and evident overlap is observed between the fluorescence profiles of PRDX-2 and dyskerin.

**Figure 3 genes-14-01766-f003:**
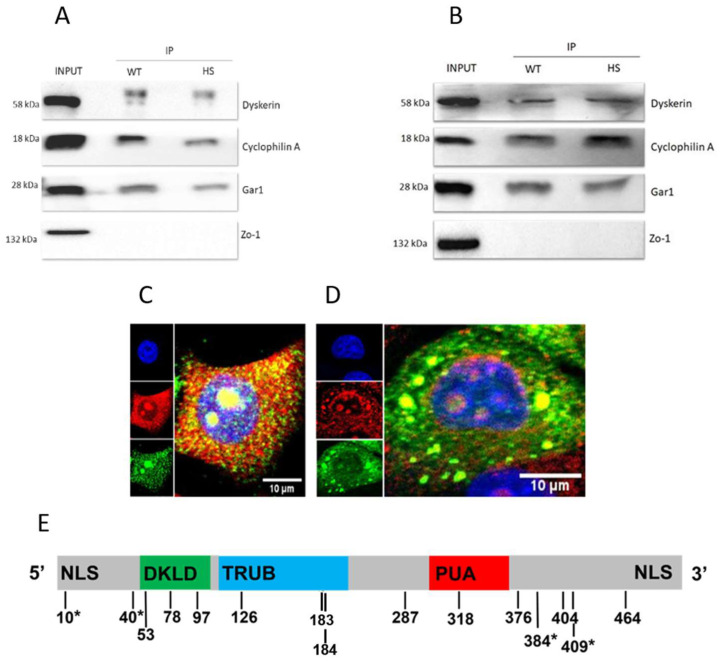
CypA-dyskerin interfacing. (**A**,**B**) Pull-down experiment supporting CypA-dyskerin interaction. U2OS (**A**) and HEK (**B**) cells were transfected with the 3xFLAG-dyskerin plasmid and the lysates incubated with the ANTI-FLAG M2 Affinity Gel beads, as described in Methods. Pulled-down complex was analyzed by immunoblotting; transfected dyskerin was detected by the anti-FLAG antibody, CypA, GAR1, and Zo-1 proteins by their specific antibodies. Abbreviations: immunoprecipitation (IP); total extract (INPUT); wild-type (WT); and heat shock (HS). (**C**,**D**) Confocal immunofluorescence pictures of 3xF-Iso3 cells (that overexpress the Iso3 dyskerin splice variant) show Iso3 and CypA co-localization in absence (**C**) or presence (**D**) of oxidative stress. Iso3 (in green) is labelled by FLAG antibody, CypA (in red) by its specific antibody. Nuclei were counterstained with DAPI (blue). Scale bars correspond to 10 μm. (**C**) Note the overlay of Iso3 and Cypa immunosignals (in yellow) at both the nuclear and cytoplasmic compartments. (**D**) Sodium arsenite exposure triggers Iso3 recruitement at SGs and leads CypA to similarly assume a cytoplasmic granular pattern, with its immunosignal labelling a subset of Iso3-positive cytoplasmic foci. (**E**) Schematic representation of dyskerin sequence and its functional domains; the position of proline residues that might potentially be targeted by CypA PPLase is indicated below. The prolines whose missense substitutions proved to be X-DC pathogenic (Pro10Leu, Pro40Arg, Pro384Ser, Pro384Leu, Pro409Leu, Pro409Arg) are marked by asterisks.

**Figure 4 genes-14-01766-f004:**
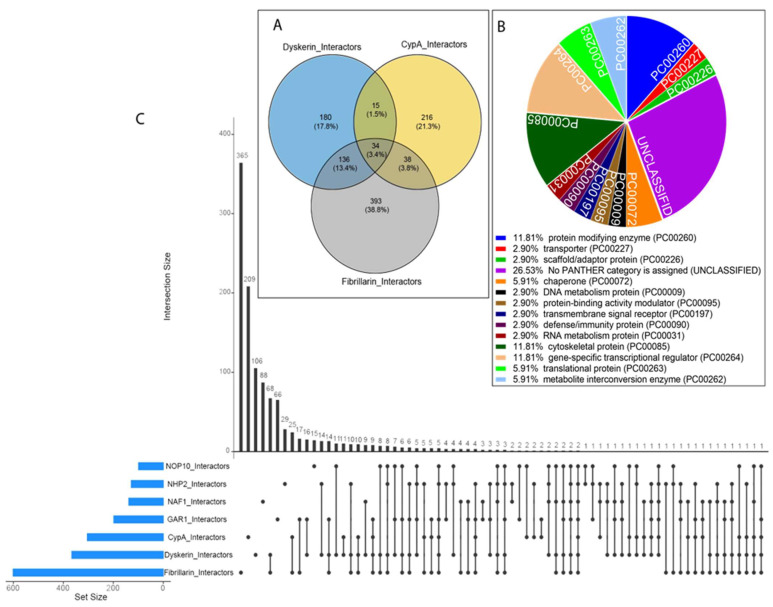
Overlay between CypA’s interactome and those of core components of nucleolar snoRNPs. (**A**) Percentage of components of the dyskerin, fibrillarin, and CypA interactomes. (**B**) Interactors common to CypA, fibrillarin, and dyskerin clustered in protein classes according to PANTHER classification system (http://www.pantherdb.org (accessed on 7 July 2023)). (**C**) Relationship between interactors of dyskerin, CypA, NAF1, Fibrillarin, Nop10, NHP2, and Gar1: an UpSet plot analysis (Appendix A). The plot provides an analysis of the relationships among the interactors associated with dyskerin, CypA, NAF1, Fibrillarin, Nop10, NHP2, and Gar1, offering a comprehensive view of the data’s intersection patterns and revealing how these interactors are interconnected. The specific lists of interactors were obtained from BioGrid (https://thebiogrid.org (accessed on 7 July 2023)).The plot includes horizontal blue bars representing the sets and the size of the compared groups. Intersecting lines or “intersections” connect the sets, indicating the presence of interactors shared between the corresponding groups. A single dot without any intersection line represents interactors unique to a specific group. The height of the vertical column corresponds to the number of interactors in a specific group, which is indicated by a dot above the column. The height also represents the number of interactors shared among two or more groups, which is indicated by an intersecting line above the column.

**Table 1 genes-14-01766-t001:** Function of a subset of CypA, dyskerin, and fibrillarin common interactors and their related diseases. Protein class (PC) accession number derived from the site http://www.panther.org (accessed on 7 July 2023). For proteins marked by the asterisk, PC was unavailable at that site and functional classification is in accordance with the uniprot.org site or the https://www.ncbi.nlm.nih.gov/gene (accessed on 7 July 2023) site. Associated diseases were derived from the www.genecards.org (accessed on 7 July 2023) site.

Protein Name	Function	Associated Disease
ARRB2	Transmembrane receptor regulatory/adaptor protein	Synovium Neoplasm; Cecum Lymphoma
BIRC3	Enzyme Modulator (PC00095)	Lymphoma, Mucosa-Associated; Lymphoid Type Lymphoma
BTF3	Transcription Factor (PC00090)	Ciliary Dyskinesia
CIT	Non-Receptor Serine/Threonine Protein Kinase (PC00167)	Microcephaly 17, Primary, Autosomal; Primary Autosomal; Recessive Microcephaly
COX15	Chaperone (PC00072)	Mitochondrial Complex IV Deficiency, Nuclear Type 6; Leig Syndrome with Leukodystrophy
CUL3	Scaffold/Adaptor Protein (PC0026)	Pseudohypoaldosteronism, Type IIe; Neurodevelopmental Disorder, With or Without Autism or Seizures
CUL7	Scaffold/Adaptor Protein (PC00226)	Three M Syndrome 1; Shox-Related Short Stature
DDRGK1	Ubiquitin-Protein Ligase (PC00234)	Spondyloepimetaphyseal Dysplasia Shohat Type; Spondyloepimetaphyseal Dysplasia
DLST	Transferase (PC00220)	Paragangliomas 7 and Hereditary Paraganglioma–Pheochromocytoma Syndromes
EFTUD2	Translation Elongation Factor (PC00222)	Mandibulofacial Dysostosis, Guion–Almeida Type; Esophageal Atresia
FANCD2 *	DNA Repair, Chromosomal Stability	Fanconi Anemia, Complementation Group D2 and Fanconi Anemia, Complementation Group A
G3BP1	RNA Metabolism Protein (PC00156)	Chikungunya and Mouth Disease
HSD17B10	Oxidoreductase (PC00176)	Hsd10 Mitochondrial Disease and Syndromic X-Linked Intellectual Disability Type 10
KIF14	Microtubule Binding Motor Protein (PC00156)	Meckel Syndrome 12 and Microcephaly 20, Primary, Autosomal Recessive.
KIF20A	Microtubule Binding Motor Protein (PC00156)	Cardiomyopathy, Familial Restrictive, 6; Familial Isolated Restrictive Cardiomyopathy
KIF23	Microtubule Binding Motor Protein (PC00156)	Anemia, Congenital Dyserythropoietic, Type IIIa and Anemia, Congenital Dyserythropoietic, Type Ia
MYC	Transcription Cofactor (PC00217)	Burkitt Lymphoma; High-Grade B-Cell Lymphoma Double-Hit/Triple-Hit
MYCN	Transcription Cofactor (PC00217)	Diseases associated with MYCN include Feingold Syndrome 1 and Familial Retinoblastoma
NPM1	Chaperone (PC00072)	AML With Myelodysplasia-Related Features; Leukemia, Acute Myeloid
NR2C2	C4 Zinc Finger Nuclear Receptor (PC00169)	Aging and Premature Aging
NTRK1	Transmembrane Signal Receptor (PC00197)	Insensitivity To Pain, Congenital, With Anhidrosis and Thyroid Carcinoma Familial Medullary
OBSL1 *	Cytoskeletal adaptor protein	Three M Syndrome 2; Three M Syndrome 1
PARK2 *	E3 ubiquitin-protein ligase	Parkinson Disease 2, Autosomal Recessive Juvenile and Leprosy 2
PRC1	Non-Motor Microtubule Binding Protein (PC00166)	Bladder Cancer
RC3H1 *	ROQUIN-1	Immune Dysregulation and Systemic Hyperinflammation Syndrome.
RECQL4	DNA Helicase (PC00011)	Baller–Gerold Syndrome; Rapadilino Syndrome
RPL11	Ribosomal protein (PC00202)	Diamond-Blackfan Anemia 7 and Diamond–Blackfan Anemia
SOAT1	Acyltransferase (PC00042)	Atherosclerosis Susceptibility; Adrenal Carcinoma
SRPK1 *	Protein Kinase	Denys–Drash Syndrome

## Data Availability

Data are contained within article/Appendix A.

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
