# Peer review of "New Insights into Dyskerin-CypA Interaction: Implications for X-Linked Dyskeratosis Congenita and Beyond"

_genes, 2023, doi:10.3390/genes14091766_

Round 1

Reviewer 1 Report

In this manuscript, Belli, V et al. attempt to clarify the role of cyclophilin A (CypA) in the nucleus and suggest that CypA affects the activity of all ribonucleoproteins by interacting with the pseudo-uridine synthase dyskerin and modulating its activity. Although there are many speculations, and some assumptions and speculations are a bit risky, this is a thought-provoking and very interesting paper. It will contribute greatly to the study of chaperone molecules and the nucleolus, as well as RNPs. However, the manuscript should be revised as it currently has several concerns. The authors should consider the comments and improve the manuscript. 

There are not so serious comments, but there are a few areas where it would be better to redo the experiment and make the data of higher quality. Therefore, I recommend this manuscript for “major revision”.

Major comments

1.  Figure 1, especially Figure 1C, lacks detail because the antibody concentration may be too high or the exposure may be too long, the signal is too strong and the image is blurry. It should be a clearer picture, like Figure 2A.

2. In addition, in Fig. 1C, HeLa cells show a distinctly different localization of CypA from RKO cells and U2OS cells. Why? At least in the Discussion, it should be inferred by consideration.

3. On line 215, the authors state that dyskerin is also concentrated in Cajal bodies. Why can the authors conclude that white ↪️ is Cajal bodies? It should be confirmed by using antibodies specific for Cajal bodies, for example.

4. From line 223, the authors state that based on the different intensity at which the CypA immunosignal was simultaneously detected at CBs and nucleoli in the same cells, the nucleoli should be expected to represent the major sites where CypA plays a more relevant role. Why can they say that?

5.  Prolines, indicated by asterisks in Fig. 3D, are scattered in various domains of dyskerin, are they all targets of CypA? In addition, although it is generally believed that enzymes leave their substrates after catalyzing them, the experimental results suggest that they are co-localized. Why?

6.  Related to 5, given the results for Iso3, could CypA translocate to the nucleus by interacting with dyskerin because dyskerin has NLS?

7. In Table 1, the authors state that the dysfunction of almost all these common partners, despite their variegated functional roles, was invariantly related to diseases. Is there a commonality in the pathology of these diseases?

8.  The authors conclude in the abstract that the data presented here suggest that this chaperone can modulate dyskerin activity influencing the activity of all its participated RNPs. However, there are no experimental results on the activity of dyskerin.

Minor comments

1. The text in Figure 4 is too small to be legible without magnification.

2. The numbers on lines 403 and 418,419, but the decimal point is usually a period, not a comma.

Reviewer 2 Report

The manuscript titled "New Insights into Dyskerin-CypA Interaction: Implications for 2 X-Linked Dyskeratosis Congenita and Beyond" by Belli et al describes the interaction of Cyclophilin A (CypA) and Dyskerin and their association with X-linked dyskeratosis (X-DC) and Hoyeraal-Hreidarsson congenital ribosomopathies. The authors investigate this with a series of Immunofluorescence experiments and LC MS/MS-based proteomics. 

1. The MS-analysis must be descbried in detail in methods as well as in the results. Its rationale should also be mentioned in the introduction. Further, a section on how was the MS/MS data analyzed and which quantifaction method was approached (was it label free)?  Overall, the proteomic experiment and its details should be described in detail. Also the data (raw  and result files) must be made availale at PRIDE repository. 

2. The white arrows in the figure 2 is not shown/unclear. 

3. What is '00' under protein name in table 1? 

4. The gene names must be italicized throughout the manuscript (eg line 119).

Moderate editing of English language required

Reviewer 3 Report

This work concerns the identification of partners of the cyclophilin A protein and its potential application for influencing ribosomopathies. The authors carefully performed the work by in vitro and in vitro methods. They showed CypA interacts with the pseudouridine synthase dyskerin and is involved in the assembly and trafficking of heterogeneous ribonucleoproteins. The identification of the CypA-dyskerin interaction showed the involvement of CypA in the modulation dyskerin activity, in particular its proline mutants. These data may be important for the study of X-linked dyskeratosis.

The results obtained are really interesting and worthy of publication. The experiments were done neatly and well presented in the manuscript. However, when considering the captions to some figures, questions arise that require clarification.

Figure 1A - for the RKO cells, the blue image (cell nuclei) is located below, although for the other two cell lines they are located at the top.

Figure 1, legend. The end first line. The cells lines called "Hella@ instead of HeLa.

Figure 2. There are no white arrows on the images. Also, the dashed rectangle is too thin. It should be more visible.

Figure 3A. I would like an explanation of the abbreviations at the top of the figure. What does INPUT, IP, HS mean? I can guess what that means, but a precise explanation in the caption is needed.

Figure 3, legend. (B) Cypa - misprint. Again, I cant found any white arrows. 

As far as I can tell, the quality of the English is quite good.

Round 2

Reviewer 1 Report

The revised manuscript has been improved by addressing the comments seriously. Although some of the figures are not very effective, I conclude that, in general, the level of detail is acceptable for publication. The authors are to be congratulated for their efforts. I therefore consider that this revised manuscript should be accepted.

Reviewer 2 Report

The authors have incorporated/addressed all the comments. However, the resolution of figures must be improved.